# Cryo-EM structure of the bifunctional secretin complex of *Thermus thermophilus*

**Edoardo D'Imprima[1], Ralf Salzer[2†‡], Ramachandra M Bhaskara[3†], Ricardo Sánchez[4], Ilona Rose[2], Lennart Kirchner[2], Gerhard Hummer[3,5], Werner Kühlbrandt[1], Janet Vonck[1*], Beate Averhoff[2*]**

[1]Department of Structural Biology, Max Planck Institute of Biophysics, Frankfurt, Germany; [2]Molecular Microbiology and Bioenergetics, Institute of Molecular Biosciences, Goethe University Frankfurt, Frankfurt, Germany; [3]Department of Theoretical Biophysics, Max Planck Institute of Biophysics, Frankfurt, Germany; [4]Sofja Kovalevskaja Group, Max Planck Institute of Biophysics, Frankfurt, Germany; [5]Institute of Biophysics, Goethe University Frankfurt, Frankfurt, Germany

**Abstract** Secretins form multimeric channels across the outer membrane of Gram-negative bacteria that mediate the import or export of substrates and/or extrusion of type IV pili. The secretin complex of *Thermus thermophilus* is an oligomer of the 757-residue PilQ protein, essential for DNA uptake and pilus extrusion. Here, we present the cryo-EM structure of this bifunctional complex at a resolution of ~7 Å using a new reconstruction protocol. Thirteen protomers form a large periplasmic domain of six stacked rings and a secretin domain in the outer membrane. A homology model of the PilQ protein was fitted into the cryo-EM map. A crown-like structure outside the outer membrane capping the secretin was found not to be part of PilQ. Mutations in the secretin domain disrupted the crown and abolished DNA uptake, suggesting a central role of the crown in natural transformation.

DOI: https://doi.org/10.7554/eLife.30483.001

**\*For correspondence:**
janet.vonck@biophys.mpg.de (JV);
averhoff@bio.uni-frankfurt.de (BA)

[†]These authors contributed equally to this work

**Present address:** [‡]Structural Studies Division, Medical Research Council – Laboratory of Molecular Biology, Cambridge, United Kingdom

## Introduction

Natural transformation is a major mode of horizontal gene transfer (*Blokesch, 2017*; *Mell and Redfield, 2014*), by which bacteria take up DNA directly from their environment. In this way, bacteria gain novel genetic information, for example metabolic traits, pathogenicity determinants and resistance genes as a driving force for bacterial adaptation and evolution. By enabling pathogenic bacteria to adapt to the human host, natural transformation is clinically relevant.

Natural transformation is a complex process mediated by multi-component transport machineries consisting of so-called competence proteins, which are highly conserved throughout the bacterial world (*Averhoff, 2009*; *Chen et al., 2005*; *Koomey, 1998*). In Gram-negative bacteria this machinery spans the entire periplasm and connects the inner (IM) and outer membranes (OM), mediating DNA binding on the cell surface and subsequent translocation through the periplasm into the cell.

Many conserved proteins of the bacterial DNA uptake machineries are similar to components of protein secretion and type IV pilus biogenesis systems and often play a dual role (*Hobbs and Mattick, 1993*). Members of the secretin protein family are conserved key components of natural transformation systems in Gram-negative bacteria, and also play important roles in protein secretion, type IV pilus extrusion and the assembly and extrusion of filamentous bacteriophages (*Ayers et al., 2010*; *Korotkov et al., 2011*). The OM-embedded C-terminal secretin domain likely provides an aperture for DNA and protein translocation through the OM, connecting the periplasm to the

extracellular environment. The members of the secretin family form homo-oligomers consisting of 12 to 19 subunits surrounding a central channel (*Bayan et al., 2006*; *Costa et al., 2015*; *Jain et al., 2011*). The size of the complex, its oligomeric state and the structures of its periplasmic N-terminal domains vary considerably between organisms. Koo et al. (*Koo et al., 2016*) showed the overall structure of the secretin domain of PilQ from *Pseudomonas aeruginosa* at intermediate resolution. Moreover, atomic models from cryo-EM structures of type 2 (T2SS) (*Yan et al., 2017*) and type 3 (T3SS) secretion systems (*Worrall et al., 2016*) revealed the conserved architecture of the secretin domain as a double β-barrel ('barrel-in-barrel' structure) forming the inner gate and the external scaffold. The N-terminal domains of secretins consist of copies of stacked rings which form a periplasmic channel that connects to IM-associated proteins, forming a multi-component secretion system. Structural information on the N domains is scarce, and the N0 domain is usually missing or disordered (*Berry et al., 2012*; *Koo et al., 2016*; *Worrall et al., 2016*; *Yan et al., 2017*).

In previous studies we discovered a unique secretin (PilQ) complex in the thermophilic bacterium *Thermus thermophilus*, which is essential for natural transformation and extrusion of type IV pili (*Friedrich et al., 2002*; *Schwarzenlander et al., 2009*). We showed that *T. thermophilus* PilQ contains a thermostable C-terminal secretin domain and an exceptionally long N-terminal tail of six stacked rings (N0-N5) (*Burkhardt et al., 2011*; *2012*; *Salzer et al., 2016*). Furthermore, we determined the in situ structure of the entire T4PS machinery by cryo-electron tomography (cryo-ET) in the piliated and unpiliated state, revealing conformational changes of the N-terminal ring-forming domains (*Gold et al., 2015*).

We now report a 7 Å resolution cryo-EM structure of the 13-fold symmetric PilQ complex revealing unique molecular architecture. We used a novel image processing tool, *REcenter Particles* (REP), and state-of-the-art remote homology detection and modeling methods to determine the structure of the large and flexible PilQ complex. A crown-like density outside the OM was unaccounted for by the molecular model of PilQ and represents an unidentified DNA translocator component essential for the function of the secretin in natural transformation.

## Results

### Cryo-EM structure of PilQ

We used carbon-back-coated holey carbon films for cryo-EM sample preparation (*Koo et al., 2016*; *Low et al., 2014*; *Reichow et al., 2010*; *Schraidt and Marlovits, 2011*; *Tosi et al., 2014*; *Yan et al., 2017*) to ensure an even distribution of PilQ particles. By this procedure, a large majority of the particle images are side views (*Figure 1*). Top views were too rare to determine the stoichiometry of the complex reliably by two-dimensional (2D) classification. Top views were subjected to multivariate statistical analysis for an unbiased assessment of particle symmetry (see Materials and methods). This procedure indicated unambiguous 13-fold rotational symmetry of the PilQ complex (*Figure 1—figure supplement 1*).

The particle images on the grid and the 2D class averages (*Figure 1*) show the typical rod shape of *T. thermophilus* PilQ (*Burkhardt et al., 2011*; *Salzer et al., 2016*). The complex has a C-terminal secretin domain, which inserts into the OM, and six N-terminal domains that form six stacked rings N0 to N5, which were found to extend deeply into the periplasm (*Burkhardt et al., 2011*; *Gold et al., 2015*). Rings N1 to N5 consist of predicted –βαββα– folds whereas the N0 ring has an unusual –ααβαββα– fold (*Burkhardt et al., 2012*; *Salzer et al., 2016*).

Although the secretin domain resists denaturation even at high temperatures in the presence of SDS (*Burkhardt et al., 2011*), the 2D class averages show clearly that the overall complex is flexible. The distances between cap and tail and between the rings vary and the cap module shows a distinct rocking movement around an apparent hinge at the interface between rings N4 and N5 (*Figure 1I, II,III,IV*). Consequently, initial 3D reconstructions had a resolution of only ~20 Å (*Figure 2F*). To improve the map we split the structure into two modules at the N4 ring and reconstructed both parts separately by residual signal subtraction (RSS) (*Bai et al., 2015*). This resulted in a tail module, comprising rings N0-N4 and a cap module containing the secretin domain along with the N5 ring. 3D classification and auto-refinement improved the resolution of both maps to ~15 Å. Residual signal subtraction improved the map especially in the secretin domain, but the alignment accuracy was poor, due to the low signal of the subtracted particles.

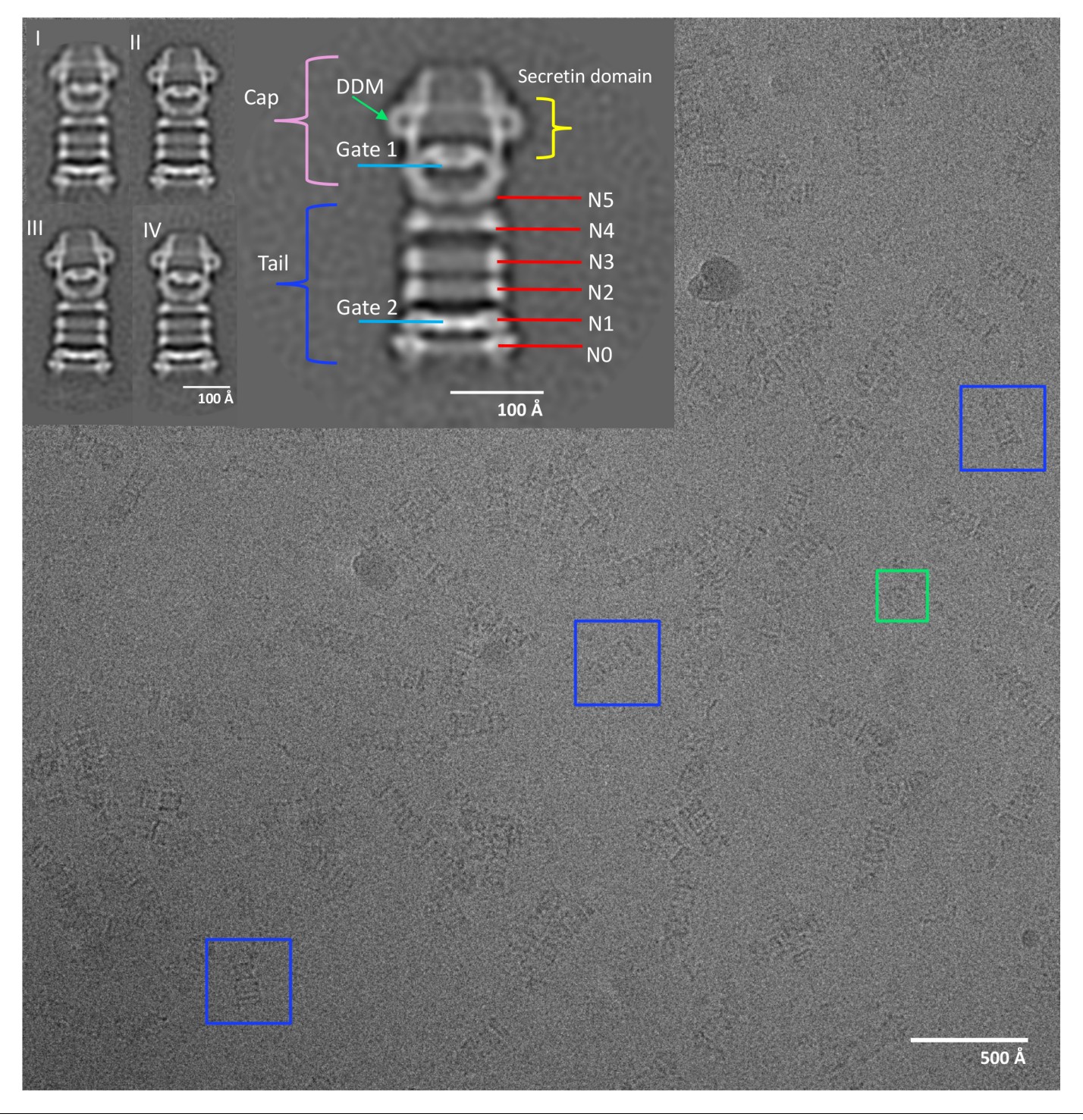

**Figure 1.** Electron micrograph of PilQ complex and selected 2D class averages. Typical micrograph of purified WT PilQ recorded at 1.8 µm defocus. Individual PilQ particles are clearly recognizable, either as frequent side-views (blue boxes) or less frequent top-views (green boxes). Insets show a selection of 2D class averages. Class I to IV were obtained by averaging 1096, 935, 728, and 691 particles respectively. Class II (2x magnification) shows the salient features of the PilQ complex with Gate 1 and 2 (blue lines), rings N5 to N0 (red lines) forming the tail (dark blue bracket) and the cap (pink bracket) containing the secretin domain (yellow bracket). Classes III and IV indicate a rocking motion of the cap and N5 ring relative to the long axis of PilQ, with a hinge at the N4 ring. Green arrow, detergent micelle (DDM).

DOI: https://doi.org/10.7554/eLife.30483.002

The following figure supplement is available for figure 1:

*Figure 1 continued on next page*

*Figure 1 continued*

**Figure supplement 1.** Assessment of PilQ symmetry and oligomeric state.

DOI: https://doi.org/10.7554/eLife.30483.003

To improve the map further, we wrote the program *REcenter Particles (REP)* (*Sanchez, 2017*) based on the approach of *Ilca et al. (2015)* and *Hou et al., 2017* (see the Appendix for a detailed description of the program). By this procedure the residual signal can be moved to any desired position in the box to maximize alignment precision, and the windows can be cropped to a smaller box size.

We applied the *REP* procedure to create four new particle data sets, one for each of the rigid parts of the PilQ structure, i.e. the secretin domain with the N5 ring, the N4 ring, and the N2N3 and N0N1 ring pairs (*Figure 2B–E,H*). The new stacks of re-centered particles were reprocessed using featureless spheres and cylinders as initial references for the secretin-N5 ring and ring modules. This improved the accuracy of the rotational alignment from ~12° to 2°, which doubled the resolution and revealed a wealth of new, interpretable map features (*Figure 2* and *Figure 2—figure supplement 1*). The final component maps have respective resolutions of 7.6 Å, 6.7 Å, 7.6 Å, 7.0 Å for the secretin-N5 ring, N4 ring, and the N2N3 and N0N1 ring pairs (*Figure 2—figure supplement 1*).

## The PilQ model

The higher-resolution component maps provided us with precise information on the position of each domain in the PilQ complex. By remote homology detection (see Materials and methods) we were able to find and align homologous 3D structures for all PilQ domains (*Table 1*; *Figure 3—figure supplement 1*). This provided us with starting templates and allowed us to build atomic models of the individual PilQ domains. $C_{13}$ symmetry was imposed on the ring domains and the symmetrized models were fitted to their map regions, followed by optimization with a flexible fitting procedure (see Materials and methods; *Table 1*). The linkers connecting the domains were then modeled and refined in a final round of molecular dynamics flexible fitting (MDFF) (*Trabuco et al., 2008*), resulting in a full-length atomic model of the PilQ complex (*Figure 3B–D*). The C-terminal secretin domain of PilQ was modeled on the basis of the recently determined high-resolution cryo-EM structures of type 2 (*Yan et al., 2017*) and type 3 secretion systems (*Worrall et al., 2016*) and fills a substantial proportion of the secretin-N5 region. The remainder of the PilQ complex is formed by six rings that consist primarily of the N-domains. The rings of domains N1 to N5 show the conserved -βαββα- motif, typical of the N-terminal part of the secretion systems (*Video 1*). Domains N0 and N1 of *T. thermophilus* PilQ differ from other known N0 and N1 domain structures (*Berry et al., 2012*; *Spreter et al., 2009*) by long insertions at positions where there are loops in the template structures (*Figure 3—figure supplement 1*). The accuracy of this part is therefore limited by the quality of the sequence alignment for the central scaffold domains N0 and N1 (*Figure 3—figure supplement 1*). Overall the independently reconstructed ring modules matched the corresponding $C_{13}$ atomic models reasonably well, as shown by computation of local cross-correlation (See Materials and methods; *Figure 3E*). The initial best fitted ring models are structurally close to the MDFF refined models (with small RMSD, *Table 2*).

Densities corresponding to linker regions between domains N0 and N1, N2 and N3, and N5 and secretin were resolved within the four independent, recentered EM maps (*Figure 2I,H*). This made it possible to model the linker segments in a straightforward manner after an individual fit of the protomer domains (*Figure 3—figure supplement 1*). Densities for loops connecting N3 to N4 and N4 to N5 were not resolved. These linker segments were therefore modeled based on the arrangement of rings and their connections as seen in the homologous *V. cholerae* GspD (*Video 2*) (see Materials and methods).

## Molecular architecture of the PilQ complex

The PilQ complex is 350 Å long and 125 Å wide (*Figure 3A*). Thirteen PilQ protomers assemble to form a cylinder with a central channel that connects the outer membrane space with the periplasm. The DDM detergent micelle at the top of the secretin domain (*Figure 1I–IV*) identifies the membrane-embedded region of PilQ (*Figure 3—figure supplement 1*). The conserved secretin domain

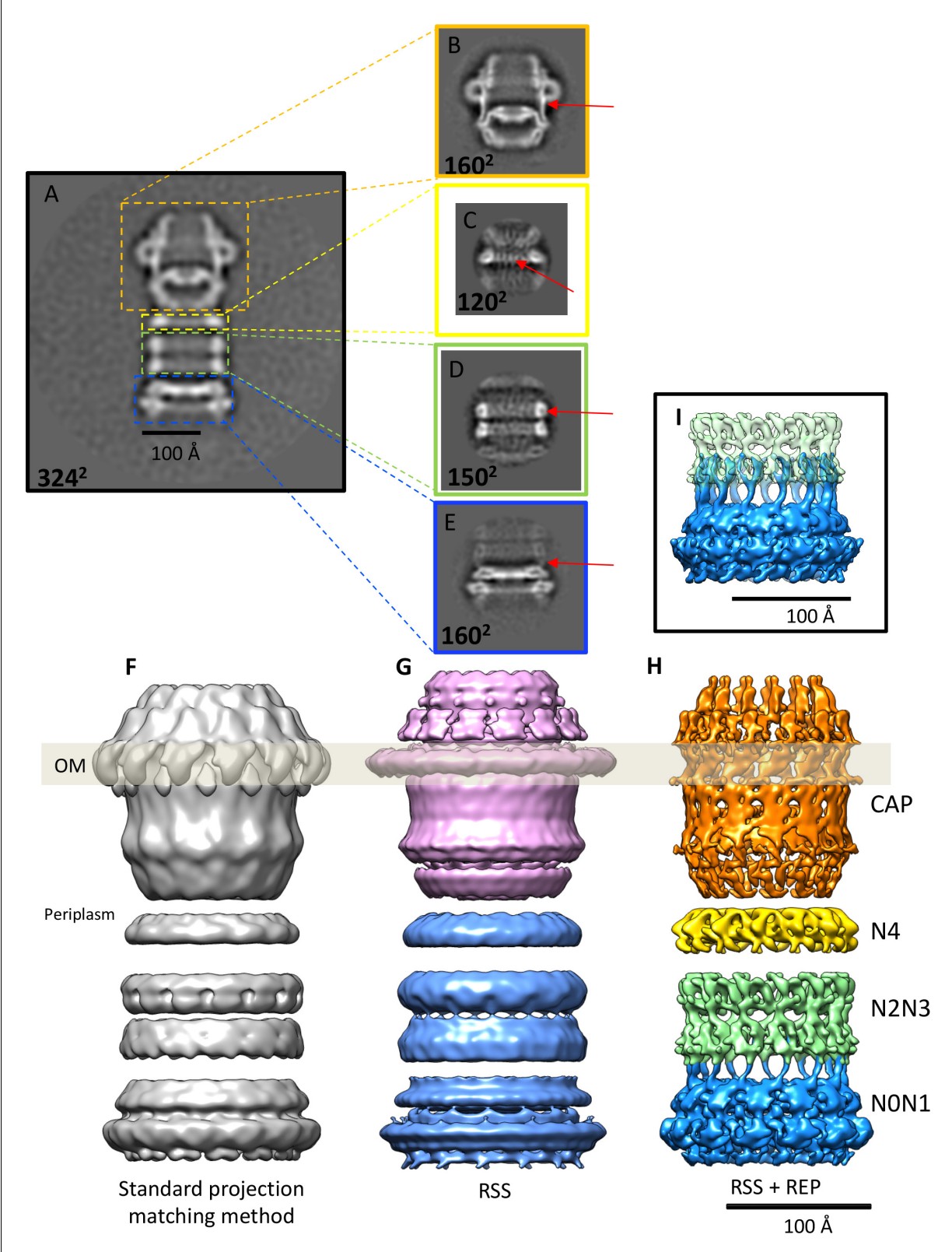

**Figure 2.** Effects of recentering in 2D and 3D reconstruction. (**A**) Typical 2D class average of 600 PilQ particles. (**B–E**) Reference-free 2D classification of the same particle subset after residual signal subtraction (RSS) and recentering, which enhances high-resolution features (red arrows) in the cap (**B**), N4 ring (**C**), N2N3 ring pair (**D**), and the N0N1 ring pair (**E**). (**F–H**) The improvement upon applying a combination of RSS and recentering is evident after 3D refinement. (**F**) Standard reconstruction yields a map of ~20 Å resolution. (**G**) Reconstructing the cap module (pink) and tail module (blue) separately

*Figure 2 continued on next page*

*Figure 2 continued*

using the RSS procedure improves the resolution to 15 Å. (**H**) Combination of RSS and recentering after dividing the PilQ complex into four modules doubled the resolution to 7 Å. Images were realigned, classified and refined independently for the cap and N5 ring (1763 particles, orange), N4 ring (3888 particles, yellow), N2N3 ring pair (1585 particles, light green) and N0N1 ring pair (2685 particles, blue). Secondary structure features were resolved in all component maps, revealing the single-polypeptide chain connections between the N0N1 and N2N3 ring pairs. (**I**) The recentering procedure does not introduce overfitting, as independently aligned and refined modules (N0N1 and N2N3) display consistent features. Features of the N2 ring along with N1-N2 linker region from both modules (blue and green) show maximal overlap. The position of the outer membrane (OM) is indicated by a transparent grey bar.

DOI: https://doi.org/10.7554/eLife.30483.004

The following figure supplement is available for figure 2:

**Figure supplement 1.** FSC curves and refinement statistics.

DOI: https://doi.org/10.7554/eLife.30483.005

---

has two distinct features: an outer β-barrel cage consisting of four long β-strands per protomer, and two β-hairpins that constitute gate 1 (*Figure 3*; *Figure 3—figure supplement 2*), as observed in the other secretin domains (*Koo et al., 2016*; *Worrall et al., 2016*; *Yan et al., 2017*) (*Figure 4—figure supplement 1*).

Sequence alignment to secretins of known 3D structure indicated that PilQ lacks the lip domain that forms a cap on the extracellular side of the OM in some species (*Yan et al., 2017*). This feature is present in *V. cholerae* GspD, while it is absent in other characterized secretion systems like *S. enterica* InvG, *E. coli* K12 GspD and *K. oxytoca* PulD (*Worrall et al., 2016*) (*Figure 4*). Surprisingly, the PilQ complex has an additional density outside the OM, which we refer to as the 'crown' (*Figure 3*, *Figure 5A*, *Video 1*). The crown consists of 13 distinct spikes, ~40 Å high, which make contact with the inner surface of the outer β-barrel of the transmembrane part of the secretin.

The top of the secretin domain has a ~70 Å aperture, which leads to Gate 1 (*Figure 3—figure supplement 2A*). Gate 1 is located immediately below the membrane-embedded portion of the β-barrel and defines a ~30 Å aperture. The aperture of Gate 1 is surrounded by multiple lever-like features that emerge from the inner barrel of the secretin domain (*Figure 3—figure supplement 2B*, *Video 1*). The N5 ring below Gate 1 has an inner diameter of ~70 Å. The N4 ring is situated below

---

**Table 1.** Modelling and fitting of PilQ domains.

Sequences of PilQ domains aligned to their respective templates were used to create atomic models of each domain. Models with the best scores (lower DOPE scores [*Sali and Blundell, 1993*]) were fitted into the corresponding density maps and screened for best fit orientations (highest cross correlation coefficient).

| PilQ Domain | Size (aa) | Consensus Secondary structural motif | Template protein (PDB code) | Sequence identity/ similarity | Resolution (Å)/Method | DOPE score (z-score) |
|---|---|---|---|---|---|---|
| N0 | 116 | βαββαββα | N0 domain of PilQ from *N. meningitidis* 4ARO_A | 11.45/ 20.61 | NA/NMR structure | −9457.03 (0.230) |
| N1 | 107 | αββββα | Periplasmic domain of secretin EscC from enteropathogenic *E. coli* 3GR5_A N1 domain model from PilQ from *N. meningitidis* 4AV2_A | 8.77/ 14.91 | 3gr5: 2.05/X-ray 4av2: NA/NMR, Model and cryo-EM | −7861.37 (0.351) |
| N2 | 57 | βαββα | Extra membrane domain of secritin HOFQ from *A. actinomycetemcomitans* 2Y3M_A | 9.72/ 20.83 | 2.3/X-ray | −5616.16 (−1.684) |
| N3 | 61 | βαββα | N domain of secretin XcpQ from *P. aeruginosa* 4E9J_A | 17.28/ 33.33 | 2.03/X-ray | −5858.94 (−1.331) |
| N4 | 56 | βαββα | N domain of secretin XcpQ from *P. aeruginosa* 4E9J_A | 14.28/ 29.87 | 2.03/X-ray | −5261.98 (−1.392) |
| N5 | 82 | βαββα | N1 domain model from PilQ from *N. meningitidis* 4AV2_A | 18.88/ 27.77 | NA/NMR, Model and cryo-EM | −6353.61 (−1.415) |
| Secretin | 230 | ββββββββαβ | Secretin domain of T2SS GspD from *V. cholerae* 5WQ8_A | 8.89/ 14.18 | 3.26/cryo-EM | −17222.63 (0.777) |

DOI: https://doi.org/10.7554/eLife.30483.012

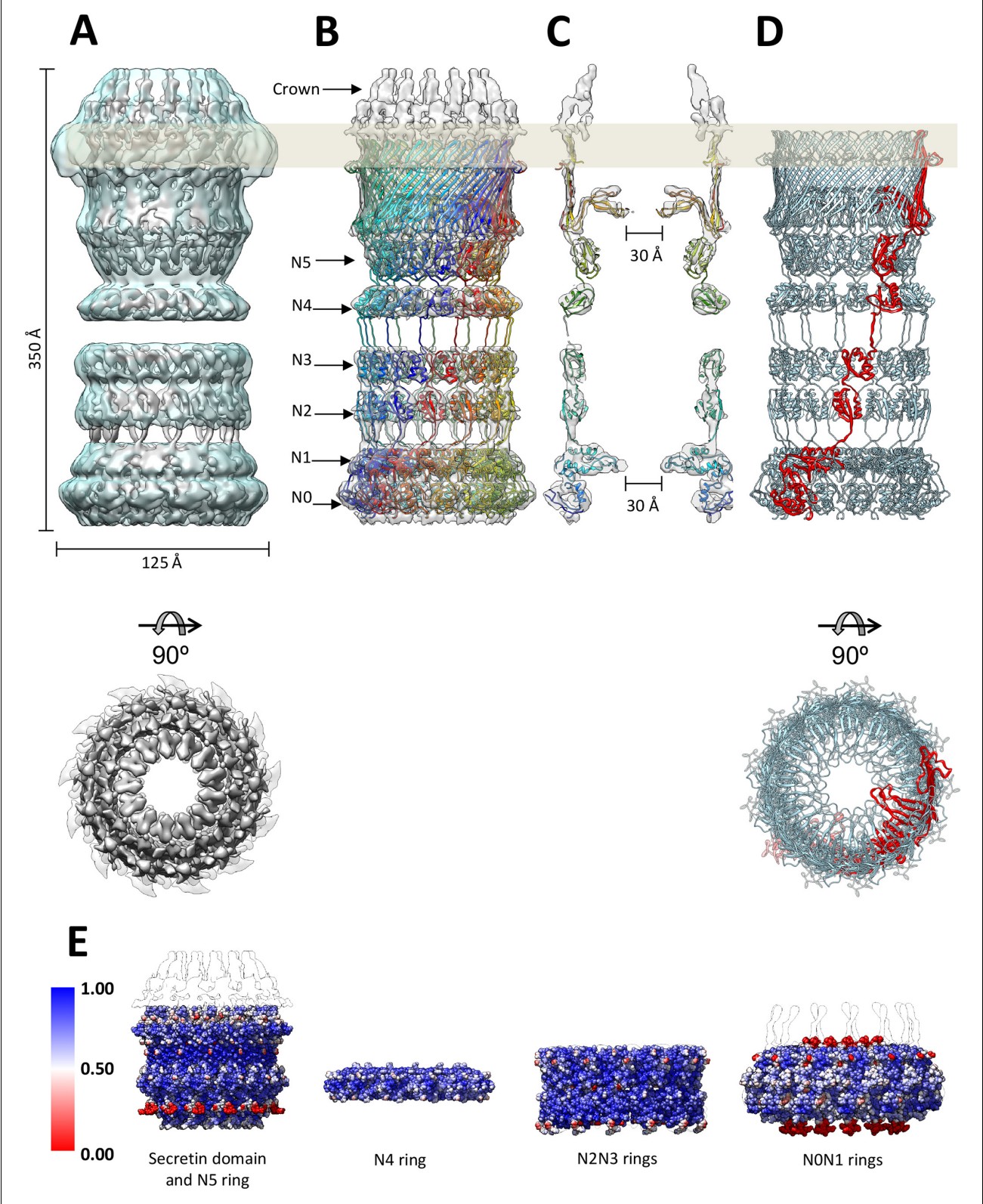

**Figure 3.** Fit of the PilQ model. (**A**) Independent reconstructions of the four modules after recentering combined into a single high-resolution map (grey), as indicated by the low-resolution envelope of the entire complex (transparent light blue). (**B**) Atomic model of $C_{13}$ symmetric PilQ complex (rainbow colored by chain) fitted into maps of individual modules after MDFF refinement (***Trabuco et al., 2008***). (**C**) Cross section of (**B**), showing the position of gates 1 and 2 relative to the outer membrane (grey bar) and the overall fit of individual domains into each ring. The density of the crown

*Figure 3 continued on next page*

*Figure 3 continued*

region outside the outer membrane is not accounted for by the PilQ sequence. (**D**) Top and side views indicating the helical arrangement of PilQ protomers (one protomer red) around the central symmetry axis. (**E**) Per voxel local cross-correlation values (CC) of $C_{13}$ symmetric model-derived maps and REP-reconstructed EM maps (outlines) were computed and used to color (red-white-blue) the atomic models (shown as spheres) before MDFF to highlight model/map correspondence.

DOI: https://doi.org/10.7554/eLife.30483.006

The following figure supplements are available for figure 3:

**Figure supplement 1.** Sequence alignments of PilQ domains.
DOI: https://doi.org/10.7554/eLife.30483.007
**Figure supplement 2.** Gates of PilQ complex.
DOI: https://doi.org/10.7554/eLife.30483.008
**Figure supplement 3.** Alternative linker models connecting N3 and N4 rings.
DOI: https://doi.org/10.7554/eLife.30483.009
**Figure supplement 4.** Assessment of the PilQ domain fits.
DOI: https://doi.org/10.7554/eLife.30483.010
**Figure supplement 5.** Model refinement.
DOI: https://doi.org/10.7554/eLife.30483.011

the N5 ring; 2D class averages indicate considerable flexibility in this region (*Figure 1I*-IV). Ring N4 is separated by ~20 Å from the N2N3 pair. The connection between this ring pair and N4 is not resolved. Given that the linker between domains N3 and N4 is 17 residues long, it is possible that consecutive rings are offset by one or two domains along the circumference of the ring (*Figure 3— figure supplement 3*) in either direction. The end-to-end distances measured for these alternative topologies suggest a range of 30–60 Å. A 17-residue linker could easily span this distance. Therefore, we modelled this linker segment in two alternative topologies (*Figure 3—figure supplement 3*).

The N2 and N3 rings form a tight pair (*Figure 2D,H*). Rings N1 and N0 likewise interact tightly (*Figure 2E,H*). The N1 ring harbors Gate 2, which has a 30 Å aperture similar to Gate 1 (*Figure 3— figure supplement 2B*). Gate 2 features a diaphragm-like arrangement with smaller twisted lever elements, which are best seen in top views (*Figure 3—figure supplement 2B*; *Video 1*). The extra loops in domain N1 contain several glycine residues (*Figure 3—figure supplement 2B*, *Figure 3— figure supplement 1*), which could confer the flexibility that may be required for opening and closing the aperture.

## K728 and R730 residues stabilize the crown region of PilQ

The crown region of the PilQ complex outside the outer membrane is not accounted for by the PilQ sequence (*Figure 3*). Our structural model identifies the region of the PilQ sequence that interacts with the crown (*Figure 5A*). We exchanged two positively charged residues (K728 and R730) at the interface between the detergent belt and the crown module to alanine by site-directed mutagenesis (see Materials and methods). These PilQ variants were expressed in a *pilQ* negative *T. thermophilus* mutant (Δ*pilQ:: bleo*).

EM analysis of the PilQ K728A/R730A (PilQ-KR) variant revealed that it forms PilQ complexes lacking the crown region (*Figure 5B–C*). To analyze the impact of these residues on PilQ complex stability or assembly, we harvested a *T. thermophilus* Δ*pilQ* mutant (Δ*pilQ::bleo*, negative control), a Δ*pilQ* mutant complemented with

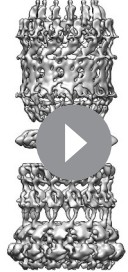

**Video 1.** Cryo-EM structure of the PilQ complex. Overall architecture of the PilQ complex. Sagittal sections along the central symmetry axis show the gates; top views of clipped planes indicate the detailed arrangement of all domains and both gates in PilQ from top to bottom.
DOI: https://doi.org/10.7554/eLife.30483.014

**Table 2.** Statistics of PilQ domain fits.

Homology models of the different PilQ domains were fitted into the respective ring module maps. Each domain was initially placed within the density maps in multiple orientations ($n = 10,000$), spanning the entire Euler angular space, and subjected to steepest descent local optimization. The resulting fits were clustered and ranked by their CC-scores with the corresponding density maps fits. The table lists the top-5 fits for each domain in the Euler-angle search, along with their CC-scores and the number of independent optimization runs ending in this cluster. The top-ranked fits were assessed for compatibility of the N-to-C-terminal domain orientations (bottom-to-top). The compatible orientations with the highest CC-value was selected as initial fit. Subsequent refinement using MDFF flexible fitting (*Trabuco et al., 2008*) further improved the fits with minimal structural change.

| PilQ domain fits | Total no. of fits | No. of unique fits | Top 5 solutions | | | Initial chosen protomer fit. CC (Cluster #) | Fit after MDFF. CC (RMSD, Å) | CC for $C_{13}$ oligomeric ring model | CC after MDFF run with masked density map |
| | | | Rank # | CC with map | Runs | | | | |
|---|---|---|---|---|---|---|---|---|---|
| N0 | 10000 | 152 | Rank 1 | 0.7028 | 576 | 0.7028 (Rank 1) | 0.7863 (3.210) | 0.70 | 0.85 |
| | | | Rank 2 | 0.6715 | 327 | | | | |
| | | | Rank 3 | 0.6696 | 167 | | | | |
| | | | Rank 4 | 0.6681 | 33 | | | | |
| | | | Rank 5 | 0.6670 | 174 | | | | |
| N1 | 10000 | 115 | Rank 1 | 0.6673 | 530 | 0.6673 (Rank 1) | 0.7682 (4.044) | 0.70 | 0.85 |
| | | | Rank 2 | 0.6639 | 682 | | | | |
| | | | Rank 3 | 0.6583 | 336 | | | | |
| | | | Rank 4 | 0.6570 | 110 | | | | |
| | | | Rank 5 | 0.6551 | 211 | | | | |
| N2 | 10000 | 48 | Rank 1 | 0.8381 | 784 | 0.8381 (Rank 1) | 0.9139 (1.883) | 0.84 | 0.91 |
| | | | Rank 2 | 0.8270 | 815 | | | | |
| | | | Rank 3 | 0.8270 | 364 | | | | |
| | | | Rank 4 | 0.8256 | 475 | | | | |
| | | | Rank 5 | 0.8248 | 953 | | | | |
| N3 | 10000 | 32 | Rank 1 | 0.8279 | 473 | 0.8249 (Rank 3) | 0.9026 (3.376) | 0.84 | 0.91 |
| | | | Rank 2 | 0.8253 | 457 | | | | |
| | | | Rank 3 | 0.8249 | 889 | | | | |
| | | | Rank 4 | 0.8237 | 829 | | | | |
| | | | Rank 5 | 0.8222 | 408 | | | | |
| N4 | 10000 | 32 | Rank 1 | 0.7982 | 1106 | 0.7982 (Rank 1) | 0.8539 (2.166) | 0.77 | 0.93 |
| | | | Rank 2 | 0.7931 | 757 | | | | |
| | | | Rank 3 | 0.7793 | 483 | | | | |
| | | | Rank 4 | 0.7521 | 434 | | | | |
| | | | Rank 5 | 0.7479 | 511 | | | | |
| N5 | 10000 | 65 | Rank 1 | 0.7394 | 206 | 0.7203 (Rank 5) | 0.7562 (2.378) | 0.77 | 0.82 |
| | | | Rank 2 | 0.7371 | 292 | | | | |
| | | | Rank 3 | 0.7274 | 364 | | | | |
| | | | Rank 4 | 0.7218 | 559 | | | | |
| | | | Rank 5 | 0.7203 | 329 | | | | |
| Secretin | 9527 | 86 | Rank 1 | 0.7634 | 672 | 0.7634 (Rank 1) | 0.8314 (1.872) | 0.77 | 0.82 |
| | | | Rank 2 | 0.7063 | 156 | | | | |
| | | | Rank 3 | 0.7013 | 285 | | | | |
| | | | Rank 4 | 0.6958 | 104 | | | | |
| | | | Rank 5 | 0.6899 | 370 | | | | |

DOI: https://doi.org/10.7554/eLife.30483.013

the pilQ$_{wt}$ gene (positive control) and a *pilQ* deletion mutant complemented with *pilQ-KR* in stationary phase and analyzed the PilQ complexes after boiling the membrane extracts for 30 min in SDS sample buffer (*Figure 6A*). No PilQ complexes were detected in the Δ*pilQ::bleo* (negative control) mutant, while the *pilQ* mutant complemented with pilQ$_{wt}$ (positive control) and *pilQ-K728A/R730A* produced assembled PilQ complexes (*Figure 6B*). We conclude that K728 and R730 are not essential for PilQ oligomerization or thermostability (*Figure 6B*), but necessary for crown assembly.

Next, we investigated the overall structural arrangement of PilQ variants. The PilQ complex formed by PilQ-KR was purified from *T. thermophilus* cells and analyzed by negative-stain EM (see Materials and methods). A total of 2911 particles were manually picked with the *boxer* module in EMAN (*Ludtke et al., 1999*) and processed for 2D reference-free classification. This procedure identified 1066 particles (~36%) without crowns (*Figure 5B*). In these particles the density close to the

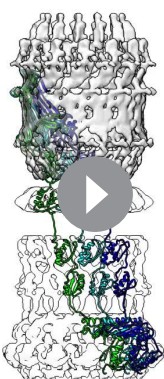

**Video 2.** Atomic model of PilQ protomer. Sagittal and longitudinal sections through the single PilQ domains show the quality of the fit of the homology models of the PilQ protomers. The overall arrangement of one PilQ protomer (chain B, cyan) after flexible fitting and linkers built and the adjacent protomers corresponding to chains A and C (blue and green respectively) show how the they are positioned relative to each other.
DOI: https://doi.org/10.7554/eLife.30483.015

OM is fuzzy (*Figure 5B*), while the remaining complex appears unchanged. This observation suggests that the crown module in this PilQ variant has bound but is not properly folded. Interestingly no perceivable differences between the *pilQ-K728A/R730A* and WT-PilQ complex were observed in the remaining 64% of the particles (*Figure 5C*). These observations illustrate that residues K728 and R730 are important for PilQ crown assembly or folding.

### K728 and R730 play a role in functionality of PilQ in twitching motility and adherence and are essential for natural transformation

To characterize the effect of PilQ-K728A/R730A mutation on the function of the PilQ complex, we quantified the twitching motility of *T. thermophilus*. The *T. thermophilus* strain containing WT PilQ-complex has a twitching zone of 2.2 cm after 3 days of incubation. This value was set to 100%. The *ΔpilQ::bleo* (negative control) was not motile (*Figure 6C*). The *pilQ-KR* double mutant showed a twitching zone of 1.4 ± 0.3 cm, which was smaller than that of the WT strain (64% of the wild-type). Next, we measured the adherence of bacterial cells to plastic surfaces, which is mediated by type-IV pili (see Materials and methods). *T. thermophilus* HB27 was used as positive control and showed a ratio (570 nm/600 nm) of 3.9 ± 1.5 × $10^{-2}$. The negative control (*ΔpilQ::bleo*) did not adhere at all. The *pilQ-KR* showed a ratio

of 1.9 ± 1.3 × $10^{-2}$. The *pilQ-KR* mutant therefore exhibited only 49% of the wild-type adherence. These findings suggest that the crown plays a role in type IV pilus-mediated adhesion and twitching motility.

To analyze the role of K728 and R730 in natural transformation, we quantified the natural transformation frequency of the *pilQ*-mutant complemented with PilQ-KR. The PilQ wildtype cells and the pilQ mutant were used as controls (see Materials and methods). The transformation frequency of the wild-type strain was 2.7 × $10^{-3}$ ± 4.9 x $10^{-4}$, while *ΔpilQ::bleo* (negative control) was not transformable. The transformability of the *pilQ-KR* mutant was reduced to 6% (frequency = 1.6 × $10^{-4}$ ± 5.4 × $10^{-5}$). This suggests that K728 and R730 play an essential, functional role in DNA uptake and natural transformation.

## Discussion

### The recentering protocol improves the resolution of flexible particles

Achieving sub-nanometer resolution for this complex required a novel tool for image processing. By applying our new recentering procedure after residual signal subtraction (*Bai et al., 2015*), we were able to align and refine each rigid module of PilQ complex independently. It also enabled us to visualize the densities of the connecting loops between some of the N-terminal ring domains (*Figure 2I*).

The overall alignment improved substantially (*Figure 2* and *Figure 2—figure supplement 1*), compensating for the small number of available particles. The improvement is easily explained as follows: A small alignment error at the center of the box becomes more severe with increasing distance from the center of rotation. Moving the particles from the edge of the box therefore results in a

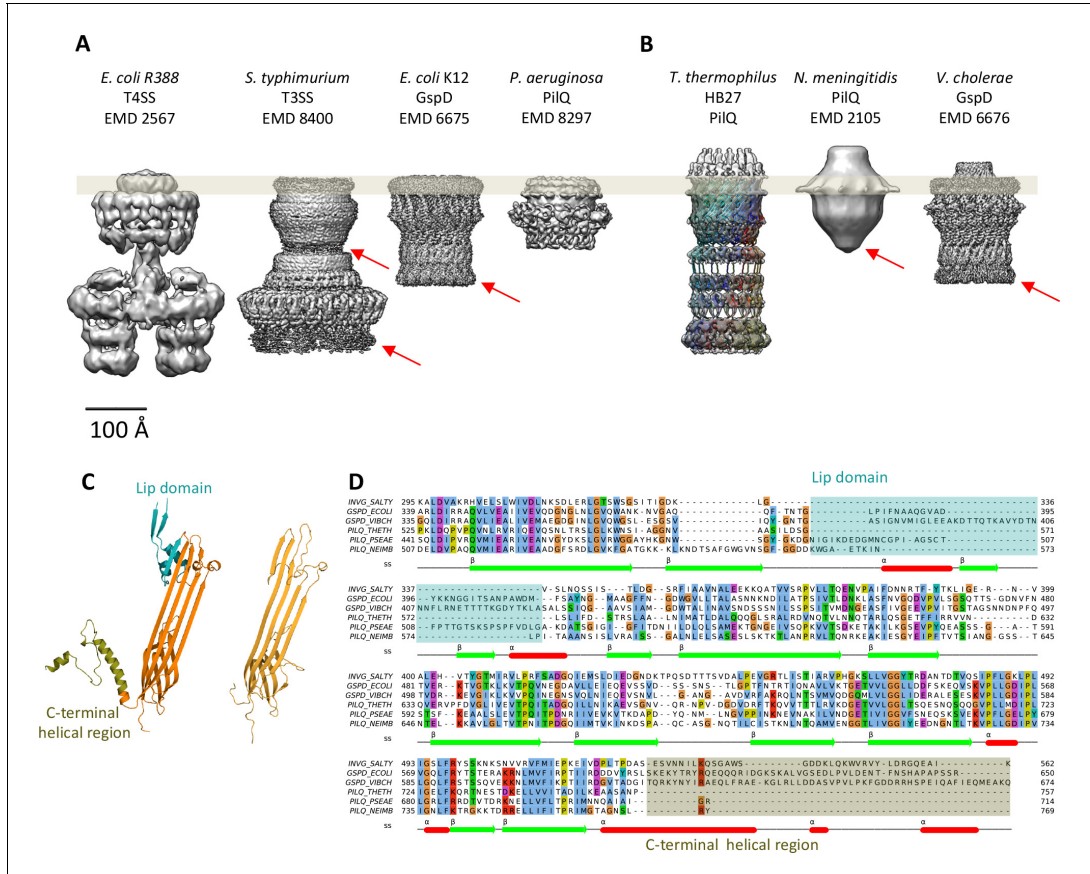

**Figure 4.** Comparison of different secretion systems with *T. thermophilus* PilQ. Cryo-EM maps of seven different secretion systems aligned against the OM (grey bar) indicate two major classes. (**A**) *E. coli* R388 T4SS, *S. enterica serovar typhimurium* T3SS (consisting of proteins InvG, PrgK and PrgH), *E. coli* K12 T2SS and *P. aeruginosa* PilQ do not have any density outside the OM (grey bar). (**B**) *T. thermophilus* PilQ, *N. meningitidis* PilQ and *V. cholerae* GspD display a conical density outside the OM. Red arrows indicate poorly resolved regions of the N-terminal domains. (**C**) The structure of the lip domain (-αβ-βα-; cyan) is resolved in *V. cholerae* GspD (left) and forms the feature outside the OM in (**B**), but it is absent in PilQ from *T. thermophilus* (right). (**D**) Multiple sequence alignment of secretin domains from well-characterized complexes indicates that the lip region (cyan) and the C-terminal helical segments (olive) are absent in all PilQ homologues. We conclude that the features outside the OM (**B**) are not formed by the PilQ protomer.
DOI: https://doi.org/10.7554/eLife.30483.016

The following figure supplement is available for figure 4:

**Figure supplement 1.** Comparison of secretin domains of different secretion systems.
DOI: https://doi.org/10.7554/eLife.30483.017

significant enhancement in the accuracy of local particle alignment. This feature of our recentering procedure should be especially useful for large, flexible complexes with multiple components such as PilQ and other secretion systems (*Worrall et al., 2016*; *Yan et al., 2017*), respiratory supercomplexes (*Sousa et al., 2016*), spliceosomes (*Nguyen et al., 2015*), apoptosomes (*Pang et al., 2015*), or stressosomes (*Marles-Wright et al., 2008*). In case of our PilQ complex, this procedure improved the map resolution to ~7 Å, whereas the conventional global alignment procedure limits the resolution to ~20 Å (*Figure 2F–H*).

The ease of implementing the recentering procedure and interfacing it with other cryo-EM software, combined with a substantial reduction in computational cost make it an attractive feature for processing cryo-EM images of large, flexible particles. Reducing the particle box size economizes on computation, particularly when using GPUs (*Figure 2—figure supplement 1*). Recentering requires only two inputs: the coordinates of the new center and the alignment parameters (see Appendix 1 for details).

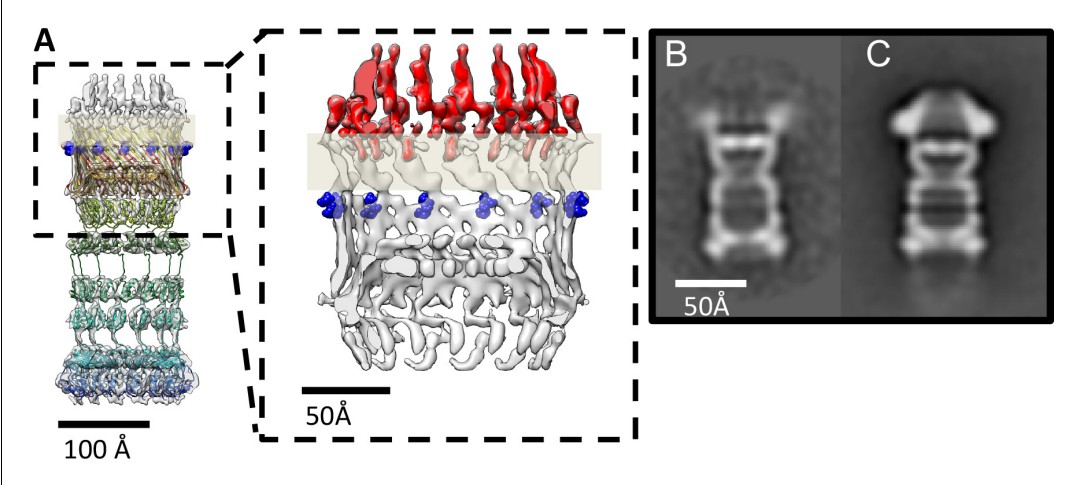

**Figure 5.** PilQ mutants K728A and R730A destabilize the crown module. (**A**) EM map of PilQ complex with fitted model (rainbow-colored from N to C terminus). Residues R730 and K728 (dark blue) are located just below the detergent belt (grey bar), close to the TM region of PilQ in the modeled homo-oligomer. Cross section through the cap region showing the crown module (red) above the OM surface. (**B and C**) Class averages of negatively stained PilQ-K728A/R730A (**B**) and wt-PilQ (**C**) obtained by averaging 374 and 3730 particles, respectively. The crown is absent in PilQ-KR, which shows weak, fuzzy densities instead, indicating that the crown is disordered.
DOI: https://doi.org/10.7554/eLife.30483.018

## Architecture of the PilQ complex

The structure of the *T. thermophilus* PilQ complex differs from that of other secretion systems (*Figure 4*). PilQ is evolutionarily related to other secretins and partly resembles T2SS (*Figure 4—figure supplement 1*). Both complexes have an OM-embedded homo-oligomeric secretin domain with a barrel-within-barrel structure (*Figure 4* and *Figure 4—figure supplement 1*).

Other known secretion systems feature a single gate in the OM (Gate 1) (*Costa et al., 2015*; *Korotkov et al., 2011*), which is a conserved feature amongst all secretins. PilQ from *T. thermophilus* has an additional gate (Gate 2) on the periplasmic side close to the platform apparatus, which connects PilQ to a still not yet identified channel in the inner membrane (*Gold et al., 2015*). Gate 2 is composed of long loops between the N0 and N1 ring (*Figure 3—figure supplement 2*). The unusual number of six N-terminal ring domains makes PilQ one of the largest secretion systems to have been characterized so far (*Figure 4*). The six rings enable the complex to span most of the wide periplasmic space of *T. thermophilus* (*Burkhardt et al., 2011*; *Burkhardt et al., 2012*). Different orientations of the individual domains and inter-protomer interactions account for slight changes in the diameter and height of individual rings. The N1 and N4 domains appear to be slightly tilted towards the inside, compared to the N2N3 pair (*Figure 3*). This inward tilting reduces the inner diameter of rings N1 and N4. In combination with the N1 inward tilt and extended loops connecting the β1-α1 and β2-β3 of the N-terminal ring, the N1 domain fills additional density within the channel-forming Gate 2 (*Video 1*, *Figure 3—figure supplement 2*). The N0 ring is well resolved in the map, whereas in other secretins it was not visible, apparently due to its flexibility (*Koo et al., 2016*; *Worrall et al., 2016*; *Yan et al., 2017*).

## Role of the crown module

Multiple sequence alignment of secretin domains (*Figure 4D*) reveals a cap/lip segment in some species that forms a protrusion outside the OM (*Koo et al., 2016*; *Worrall et al., 2016*; *Yan et al., 2017*). *T. thermophilus* PilQ does not contain this sequence; however, our EM map shows a large crown module outside the OM (*Video 1*, *Figure 3*), which cannot be assigned to any part of the PilQ sequence (*Figure 5A*) and thus must be another protein component of the complex. The crown module outside the OM is also visible in sub-tomogram averages of the piliated and the non-piliated complex in situ (*Gold et al., 2015*) (*Figure 7*). The crown module appears to be difficult to separate from the PilQ secretin domain and resists denaturation with SDS at high temperature

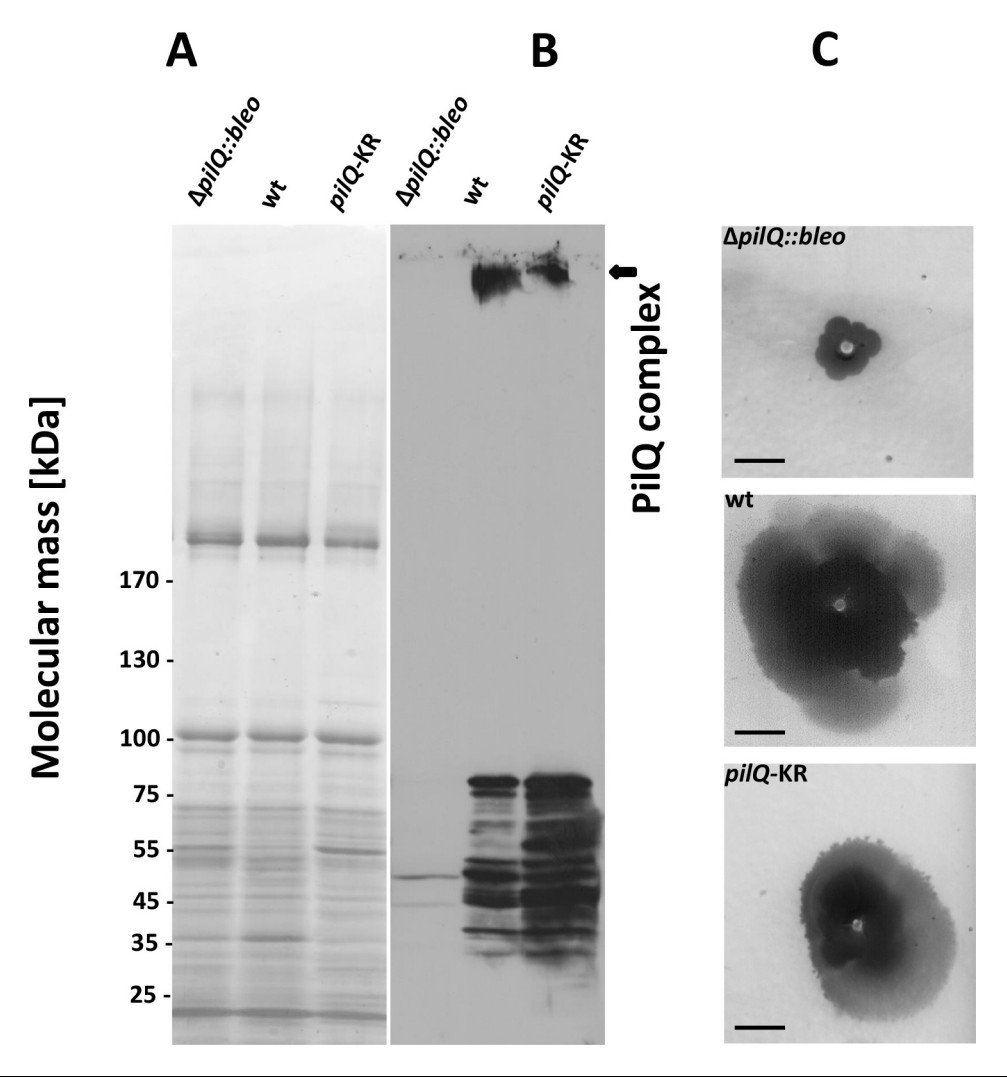

**Figure 6.** Stability of the PilQ-KR variants and twitching motility assay. (A, B) Stability of purified PilQ wildtype and PilQ-KR as indicated by SDS-PAGE (**A**) and Western blot analysis using PilQ antibodies (1:13000) (**B**) of membrane extracts boiled for 30 min in SDS sample buffer. (**C**) Contrast-inverted images of culture plates to quantify the twitching motility of the *T. thermophilus* strains (Scale bar = 5 mm).
DOI: https://doi.org/10.7554/eLife.30483.019

(*Burkhardt et al., 2011*), which indicates a very tight interaction. Negative stain EM showed that the PilQ-K728A/R730A mutation affects the structure of the PilQ complex and interferes with the crown folding or assembly (*Figure 5B–C*). This mutation has a minor effect on motility and adherence of bacterial cells. However, the transformation efficiency of these cells is severely impaired. Our previous finding that PilQ is essential for DNA binding (*Schwarzenlander et al., 2009*) leads us to conclude that the crown structure is either directly or indirectly implicated in DNA binding and uptake. We speculate that it may be the extracellular switch that converts the pilus extrusion machinery into a DNA uptake system. Experiments to identify the crown module are currently in progress.

PilQ forms the core of the bifunctional T4PS/DNA uptake system and is suggested to interact with a number of periplasmic and outer and inner membrane anchored proteins (*Friedrich et al., 2002*; *Rose et al., 2011*; *Rumszauer et al., 2006*; *Salzer et al., 2014*; *Schwarzenlander et al., 2009*; *Tsai and Tainer, 2016*). Our structure provides a firm base for mechanistic investigations of the complex apparatus that enables natural transformation.

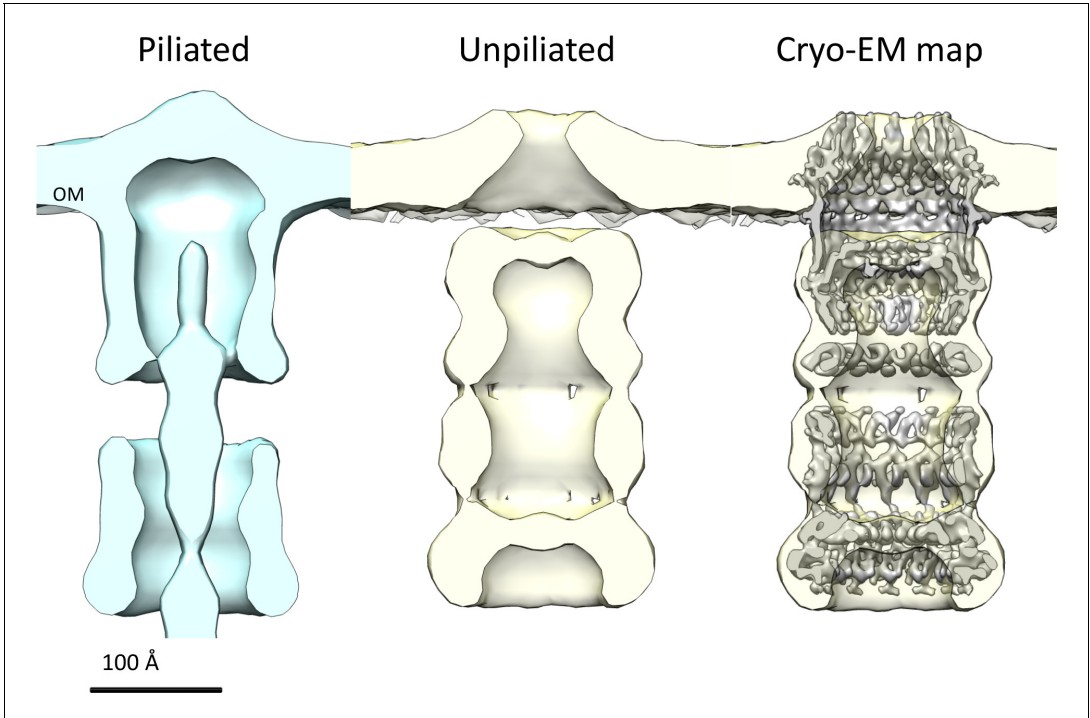

**Figure 7.** Comparison of *T. thermophilus* T4PS complex in situ to detergent-solubilized PilQ. Cross section of sub-tomogram averages of piliated (light blue) and unpiliated T4PS (light yellow) (*Gold et al., 2015*) with superposed 7 Å cryo-EM structure of PilQ complex.
DOI: https://doi.org/10.7554/eLife.30483.020

## Materials and methods

### Expression and purification of PilQ complex and variants

*Thermus thermophilus* ΔpilQ::bleo mutant cells producing his-tagged wildtype PilQ or his-tagged K728A/R730A PilQ variants encoded from the *E.coli/Thermus* shuttle vector pDM12 were grown in TM$^+$ complex (*Burkhardt et al., 2012*). Cells were harvested and 40 g cell pellet was resuspended in buffer 1 (40 mM Tris/HCl, 300 mM NaCl, 1.0 M glycine betaine, 5 mM MgCl$_2$, 5 mM CaCl$_2$, pH 8.5). The cells were disrupted by 'French Press' (3 × 1000 Psi). Membranes were prepared by ultracentrifugation (260,000 g, 4°C, 45 min), and then resuspended in buffer 1. PilQ complexes were solubilized using 3% *n*-dodecyl-ß-D-maltoside (DDM). After solubilization the supernatant was diluted to a final concentration of 1.5% DDM and incubated with 30 ml of 'TALON Metal Affinity Resin' (Clontech Laboratories, Inc., St-Germain-en-Laye, France) for 1 hr at 4°C. The column was washed with buffer 2 (40 mM Tris/HCl, 300 mM NaCl, 5 mM MgCl$_2$, 5 mM CaCl$_2$, 0.05% DDM, 12.5 mM imidazole, pH 8.5). Elution was performed with buffer 2 containing 150 mM imidazole. The elution fraction was diluted 1:2 with buffer 3 (40 mM Tris/HCl, 5 mM MgCl$_2$, 5 mM CaCl$_2$, 0.05% DDM, pH 8.5) and subjected to anion exchange chromatography (Q-Sepharose), as described recently (*Salzer et al., 2016*).

*E. coli* cultures were grown in LB medium using a similar protocol with appropriate kanamycin, ampicillin, and/or bleomycin were added. *T. thermophilus* strains were cultivated at 68°C in TM$^+$ medium to stationary growth phase and cells were harvested. Membrane extracts containing PilQ complex were prepared as described (*Salzer et al., 2016*). The membrane extracts were boiled for 30 min in SDS sample buffer and separated by 3–12% SDS-PAGE. Proteins were transferred to a nitrocellulose membrane. Western blot analysis was performed with polyclonal PilQ antibodies using a dilution of 1:13000. K728A and R730A variants of PilQ were obtained through site directed mutagenesis. pDM12-pilQ-6his plasmid (*Burkhardt et al., 2011*) along with primers, SDM_KR-for GA TCGGCGAGCTGTTCGCCCAGGCCACGAACGAGAGCACCGACAAGG and SDM_KR-rev AGGGGA TGTCCATGAGGAGGGGCACC were used. The plasmids were sequenced to confirm the mutations

and transformed into *T. thermophilus* Δ*pilQ::bleo* mutant. The resulting *pilQ* gene (*pilQK728A/R730A*) was expressed from the plasmid pDM12 under the *bc1* promoter (*Salzer et al., 2014*) in a Δ*pilQ::bleo* mutant.

## Twitching motility, adherence and transformation assay

*T. thermophilus* strains were grown for 3 days on minimal medium containing 1% BSA. The plates were subsequently stained with Coomassie Blue. *T. thermophilus* cells were detached from the agar and the twitching zones. After decantation of the cell suspension, colorless regions are visible, where the cells attach. This region (diameter in cm) is defined as the twitching zone. This is made visible as a dark zone by contrast inversion.

Adhesion of bacterial cells, which is mediated by type-IV pili, was quantified using the microtiter adhesion assay (*Burkhardt et al., 2012*). Cells were inoculated to an optical density of 0.05 (at 600 nm) and incubated for 3 days at 55°C, 64°C, or 72°C. After incubation, the optical density (600 nm) was measured again, followed by a washing step to remove non-sessile cells. Remaining adherent cells were stained with 0.1% crystal violet solution. Excess staining solution was removed by washing three times with water. Crystal violet from the adhering cells was then dissolved in ethanol, and its absorbance was measured at 570 nm. Adherence was quantified as a ratio of absorbance at 570 nm to 600 nm.

Transformation efficiency of WT and mutant cells was analyzed by performing transformations at 68°C on TM$^+$ agar medium using 5 µg of genomic DNA of a spontaneous streptomycin-resistant *T. thermophilus* HB27 mutant (*Friedrich et al., 2001*). The transformation frequency was calculated as number of transformants per living count. All transformation assays were performed in triplicate.

## Negative stain EM of PilQ-KR mutants

We used 2% ammonium molybdate for negative staining of a PilQ-KR mutant sample diluted to a final concentration of 0.01 mg/ml (*Salzer et al., 2016*). Micrographs were recorded in an FEI Tecnai G2 Spirit operated at 120 kV, at a nominal magnification of 29,000, yielding a pixel size at the specimen level of 4.2 Å. 2D classification was carried out using Xmipp 3.1 using the classification algorithm CL2D (*Sorzano et al., 2010*).

## Cryo-EM data collection

Three µl of wt PilQ sample (concentration = 1.5 mg/ml) was applied to freshly glow-discharged (at 15 mA for 25 s) Quantifoil R1.3/1.2 holey carbon grids (Quantifoil Micro Tools, Germany) back-coated with a thin carbon layer. The grids were vitrified in an FEI Vitrobot Mark IV plunge-freezer at 70% humidity and 10° C after blotting for 8–10 s. Cryo-EM images were collected in a JEOL 3200 FSC electron microscope operating at 300 kV, after coma free alignment, equipped with an in-column energy filter at a slit width of 18 eV. Images were recorded manually at a nominal magnification of 20,000x, yielding a calibrated pixel size of 1.63 Å, on a K2 direct electron detector (Gatan) operating in counting mode. Dose-fractionated 9 s movies of 45 frames were recorded with an electron dose of 0.75 e$^-$/Å$^2$/frame at a defocus of 1.2–3.4 µm.

## Cryo-EM image processing

A total of 932 micrographs was collected. Whole-image drift correction of each movie was performed using MotionCorr (*Li et al., 2013*). A second round of whole-image drift correction was performed using Unblur (*Grant and Grigorieff, 2015*) and the aligned movies were summed and exposure-filtered using the program Summovie (*Grant and Grigorieff, 2015*). The CTF was determined using CTFFIND4 (*Rohou and Grigorieff, 2015*) in the RELION workflow (*Scheres, 2012*). A dataset containing 11,457 particle images manually picked with EMAN boxer (*Ludtke et al., 1999*) (324 pixels x 324 pixels) was subjected to 2D reference-free classification in RELION (*Scheres, 2012*) to check the quality of the particle images. To assess the symmetry of the particles, 363 top-views were separately picked and extracted with a smaller box size (168 × 168). Top views were subjected to reference-free classification, but the 2D class averages did not display reliable features because of the low number of particles and the attractiveness effect of the maximum likelihood approach (*Sorzano et al., 2010*). To assess the symmetry of 2D class averages, the eigenimages were evaluated after multivariate statistical analysis (MSA) (*Böttcher et al., 2015*; *Dube et al., 1993*) in IMAGIC

(*van Heel et al., 1996*). From a class average obtained in RELION, a clear top view was selected and 50 randomly rotated copies were produced with the ROTATE-RANDOMLY command, realigned translationally, and subjected to a single round of MSA with the MSA-RUN command. This procedure was repeated with 10 other top views to assess the reproducibility. The eigenimages clearly indicated 13-fold cyclic symmetry (*Figure 1—figure supplement 1*).

Next a consensus 3D refinement was performed with all particles. As a starting reference a featureless cylinder was generated in Xmipp 3.1 (*Sorzano et al., 2010*) with the module *xmipp_transform_mask* with *–create_mask –mask cylinder* options. After the consensus refinement, the resulting map was used as a starting reference for 3D classification with local angular search. The same map was also used as a template for obtaining a soft-edge shaped mask with the *relion_mask_create* module in RELION. Several runs of 3D classification were conducted, varying the number of classes in order to assess the consistency of the results. Different symmetries ($C_9$ to $C_{18}$) were tested, but only $C_{13}$ provided consistent reconstructions in terms of interpretable features. Local angular search was performed with $1.875° \pm 10°$ and then $0.9° \pm 3°$ respectively. A dataset of 2646 particle images was obtained after combining the classes displaying the best interpretable features ,which was then subjected to the 3D auto-refinement procedure in RELION with only local angular search using the best alignment parameters obtained at the end of 3D classification. This led to two maps with a resolution of ~20 Å. To account for the observed flexibility of the PilQ complex, the map was divided into two parts: the cap, consisting of the secretin domain and the N5 ring, and the tail containing rings N0-N4. A residual signal subtraction (RSS) procedure (*Bai et al., 2015*) was applied to both parts, subtracting the signal for the other part from the aligned images, yielding two new datasets of particle images for cap and tail regions (*Figure 2*,G). Each dataset was then subjected again to a global alignment through a consensus refinement, 3D classification and auto-refinement with only local alignment as described above for the complete complex. Although some new features were partially resolved, the global alignment of the subtracted particles was poor. To resolve this problem, we used our in-house program REP (see the appendix for a detailed explanation) to re-center the residual signal of the particles within a smaller box size. This allowed us to subdivide the PilQ complex into four modules incorporating all the particles of the initial dataset with different box sizes: secretin-N5 domains ($160 \times 160$), N4 ring ($120 \times 120$), N2N3 ring pairs ($150 \times 150$) and N0N1 ring pairs ($160 \times 160$). These new datasets were reprocessed with FREALIGN (*Grigorieff, 2007*), first in a global alignment cycle (mode 3) and followed by three iterations of local alignments (mode 1) using featureless spheres and cylinders as initial references for the cap and N-terminal ring modules. Subsequently each module was subjected to 3D classification in mode 1, refining the angles every three iterations and performing the classification with 3 to 9 classes in order to test reproducibility. Alternatively, RELION 1.4 (*Scheres, 2012*) was used to accurately estimate rotational alignment accuracy as function of the box size. After post-processing, maps of the cap, N4 ring, N2N3 and N0N1 ring pairs had a resolution of 7.6, 6.7, 7.0, and 7.6 Å, respectively. Before visualization all density maps were corrected for the modulation transfer function (MTF) of the K2 direct detector. Maps were sharpened by applying a B-factor of $-250$ Å$^2$.

## PilQ modeling

To model the structure of PilQ we first identified homologous proteins with known structure by performing a BLAST search (*Altschul et al., 1990*) of the PilQ Uniprot sequence (Q72IW4) against the protein data bank (PDB). Hits were filtered by query coverage of 70% and an E-value cut-off of 3.0. The resulting template PDB structures for each of the PilQ domains were pruned by structural resolution and realigned with their respective templates using the 3D-Coffee algorithm (*Taly et al., 2011*) to maximize alignment quality and coverage. N1 and secretin domains were realigned with their respective template sequences using HHPred (*Söding et al., 2005*). These alignments were then used for generating homology models for all the individual domains using Modeller (*Sali and Blundell, 1993*; *Webb and Sali, 2014*). The modelling procedure is summarized in *Table 1*.

## Fitting homology models to the cryo-EM density map

From a series of domain deletion mutants of PilQ, we deduced that individual consecutive domains form discrete stacked ring-like density segments and are arranged with the N-terminal N0 domain at the bottom end and the C-terminal secretin domain at the top end (*Salzer et al., 2016*). This analysis

enabled us to assign all domains in the low-resolution map of the complete PilQ assembly. Further reprocessing of EM images by recentering subdomains provided us with four better-resolved density maps corresponding to (a) secretin domain and N5 domain, (b) N4 domain, (c) N3 and N2 domains, and (d) N1 and N0 domains, respectively. The homology models of individual domains were then fitted into masked ring-like densities, first as individual protomers using the *fit-in-map* tool of Chimera (*Pettersen et al., 2004*). The best-fit protomer orientations of individual domains were identified from the 15 top scoring orientations with high cross-correlation coefficients. The best models were analyzed to ensure correct arrangements of domain boundaries for each domain within the ring structures, that is, N terminus towards the base (tail) and C terminus towards the top (cap). Connecting linker segments between domains N0N1, N2N3 and, N5-secretin were modeled using the Modeller loop modeling protocol with DOPE scoring (*Sali and Blundell, 1993*) at higher precision. $C_{13}$ symmetry-related chains (A to M) corresponding to modeled protomer fragments were added to make $C_{13}$-symmetric rings of PilQ domains/fragments, followed by an exhaustive 6D rigid body search to fit into the corresponding localized density regions with an angular sampling of 20 degrees using *Colores* (*Lasker et al., 2010*). To account for the flexibility and conformational dynamics of the entire PilQ oligomeric complex, we optimized the 13-mer fragment rigid-body fits using a molecular dynamics flexible fitting (MDFF) procedure (*Trabuco et al., 2008*). MDFF adds biasing forces proportional to the observed EM density gradients. MDFF runs were set up using VMD (*Humphrey et al., 1996*) and used the CHARMM36m force field (*Huang et al., 2017*). Simulations were carried out using NAMD (*Phillips et al., 2005*) in vacuum at a temperature of 300K for 500 ps. Forces corresponding to EM density gradients were coupled using an MDFF scaling factor ξ of 0.3. The MDFF runs were followed by short energy minimization runs for 10000 steps with ξ of 10. Restraints to preserve secondary structure, chirality, and trans-peptide bond geometry were employed. The loop segments within the N0 and N1 domains are longer and more disordered than in the top-scoring template structures. To model these flexible regions, ten loop conformations were modeled and filtered to identify the model exhibiting maximum overlap with the corresponding density regions in the N0N1-masked map. Four successive MDFF runs were performed for the N0N1 domains, each for 500,000 steps, using increasing values of the scaling factor, ξ = 0.3, 0.5, 1.0, and 3.0, respectively. All MDFF runs were monitored by measuring the evolution of the backbone RMSD and the cross-correlation coefficient changes with the target density maps. To optimize the atomic model of the fully assembled complex, a complete high-resolution EM map for the entire PilQ complex was built by stacking the individually processed ring density maps, guided by a low resolution template map of the whole PilQ complex. The four optimized $C_{13}$ symmetric models were fitted into corresponding density regions, followed by modeling of missing loop segments between N1-N2 rings, N3-N4 rings, and N4-N5 rings using Modeller (*Sali and Blundell, 1993*; *Webb and Sali, 2014*) to preserve the domain connectivity as in *V. cholerae* GspD. To prevent overfitting, the $C_{13}$ symmetry-related oligomer was used for flexible fitting of the tridecameric PilQ model into the complete EM map.

## Validation of initial fits

We first generated individual domain maps of the entire PilQ protein. The initial homology domain models for all PilQ domains were then subjected to extensive sampling (n = 10000) of complete Euler angular space ($\Phi \in [0, 2\pi]$; $\theta \in [0, \pi]$; $\Psi \in [0, 2\pi]$) to obtain initial orientation ensemble for placing the respective domains within their corresponding EM maps. Following local steepest descent optimization, we obtained several unique solutions for each domain. These unique fits were characterized by their fitted-orientation, local CC and the number of runs converging into these solutions. The top best fit solutions were ranked by their local cross correlation values computed with the corresponding domain density maps using colores from the Situs package (*Lasker et al., 2010*). The previously obtained initial best fit orientations for each PilQ domain were re-evaluated by comparing them with top scoring solutions before and after MDFF refinement (*Figure 3—figure supplements 4, 5*).

## Model/map correspondence

The initial best fitted $C_{13}$ homology models before refinement were used to generate EM maps at the same corresponding nominal resolution of the recentered EM maps. Subsequently the model-

generated maps were used to compute per voxel local cross-correlation (using the *vop localCorrelation* module of UCSF Chimera) against the experimental ring-module EM maps to show model/map correspondence. This showed overall good agreement between the initial $C_{13}$ homology models and the EM maps. Regions that were not resolved in the EM maps, corresponding to linker segments and domain termini connecting individual domains, were subsequently refined using MDFF.

### Data deposition

The EM maps have been deposited in the EMDB with accession codes EMD-3985 for the full complex and EMD-3995, EMD-3996, EMD-3997 and EMD-3998 for the cap, N4, N2-N3, and N0-N1 domains, respectively. The Recentering program can be downloaded from https://github.com/rkms86/REP (a copy is archived at https://github.com/elifesciences-publications/REP).

## Acknowledgements

We thank Özkan Yildiz and Juan Castillo for computer support and Deryck J Mills and Simone Prinz for EM support. We are grateful to Martin Wilkes for discussions about sample preparation and data acquisition, to Katharina van Pee and Martin Centola for discussions about sample stabilization, and to Ahmad Reza Mehdipour for discussions on modelling and fitting. We also thank Pilar Cossio-Tejada for critical comments on the manuscript. We thank Bernd Ludwig for providing the pDM12 vector. This project was funded by the Max Planck Society and Goethe University, Frankfurt. Part of the work was supported by a grant of the German Research Foundation (DFG) Av9/6-2 to BA.

## Additional information

### Competing interests

Werner Kühlbrandt: Reviewing Editor, eLife. The other authors declare that no competing interests exist.

### Funding

| Funder | Grant reference number | Author |
| --- | --- | --- |
| Goethe-Universität Frankfurt am Main | | Beate Averhoff |
| Max-Planck-Gesellschaft | | Gerhard Hummer Werner Kühlbrandt |
| Deutsche Forschungsgemeinschaft | Av9/6-2 | Beate Averhoff |

The funders had no role in study design, data collection and interpretation, or the decision to submit the work for publication.

### Author contributions

Edoardo D'Imprima, Conceptualization, Formal analysis, Writing—original draft, Writing—review and editing, Preparation of cryo-EM samples, Devising data acquisition strategy, Acquisition of EM data, Conceptualization of REP; Ralf Salzer, Writing—original draft, performed cloning, purification and biochemical analysis, Provided samples for structure determination with help from IR and support from BA; Ramachandra M Bhaskara, Writing—original draft, Writing—review and editing, Performed modelling, Fitting and refinement of structural model with inputs from GH; Ricardo Sánchez, Wrote the REP program, Wrote the Appendix to the manuscript; Ilona Rose, Provided samples for EM with support from BA; Lennart Kirchner, Investigation; Gerhard Hummer, Supervision, Funding acquisition, Writing—review and editing; Werner Kühlbrandt, Funding acquisition, Writing—review and editing; Janet Vonck, Conceptualization, Formal analysis, Supervision, Funding acquisition, Writing—review and editing; Beate Averhoff, Conceptualization, Funding acquisition, Formal analysis, Supervision, Writing—review and editing

## Author ORCIDs

Edoardo D'Imprima (iD) https://orcid.org/0000-0002-9830-7929
Ramachandra M Bhaskara (iD) https://orcid.org/0000-0002-7742-0391
Werner Kühlbrandt (iD) http://orcid.org/0000-0002-2013-4810
Janet Vonck (iD) http://orcid.org/0000-0001-5659-8863

## Decision letter and Author response

Decision letter https://doi.org/10.7554/eLife.30483.036
Author response https://doi.org/10.7554/eLife.30483.037

## Additional files

### Supplementary files

• Transparent reporting form
DOI: https://doi.org/10.7554/eLife.30483.021

### Major datasets

The following datasets were generated:

| Author(s) | Year | Dataset title | Dataset URL | Database, license, and accessibility information |
| --- | --- | --- | --- | --- |
| D'Imprima E, Vonck J, Sanchez R | 2017 | PilQ from Thermus thermophilus | http://www.ebi.ac.uk/pdbe/entry/emdb/EMD-3985 | Publicly available at the Electron Microscopy Data Base (EMDB) (accession no: EMD-3985) |
| D'Imprima E, Vonck J, Sanchez R | 2017 | Cap domain and N5 ring of PilQ from Thermus thermophilus | http://www.ebi.ac.uk/pdbe/entry/emdb/EMD-3995 | Publicly available at the Electron Microscopy Data Base (EMDB) (accession no: EMD-3995) |
| D'Imprima E, Vonck J, Sanchez R | 2017 | N4 ring of PilQ from Thermus thermophilus | http://www.ebi.ac.uk/pdbe/entry/emdb/EMD-3996 | Publicly available at the Electron Microscopy Data Base (EMDB) (accession no: EMD-3996) |
| D'Imprima E, Vonck J, Sanchez R | 2017 | N2 and N3 ring of PilQ from Thermus thermophilus | http://www.ebi.ac.uk/pdbe/entry/emdb/EMD-3997 | Publicly available at the Electron Microscopy Data Base (EMDB) (accession no: EMD-3997) |
| D'Imprima E, Vonck J, Sanchez R | 2017 | N0 and N1 ring of PilQ from Thermus thermophilus | http://www.ebi.ac.uk/pdbe/entry/emdb/EMD-3998 | Publicly available at the Electron Microscopy Data Base (EMDB) (accession no: EMD-3998) |

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

# Appendix 1

DOI: https://doi.org/10.7554/eLife.30483.022

## Description of the Recentering program

Accurate determination of the Euler angles of each projection of a protein complex is a prerequisite for high-resolution 3D reconstruction. If the complex has different conformations or it is inherently flexible, the orientations of individual domains in a particle can be different. In that case, reconstruction of subdomains by residual signal subtraction (RSS) can be used. However, if the center of the subdomain is far away from the center of the particle box, the alignment becomes increasingly inaccurate with the distance from the center of rotation. A program for recentering (REP) was written to move the center of mass of the subdomain to coincide with the center of rotation (**Sanchez, 2017**). This makes it possible to reduce the box size of the subdomain images, which maximizes local alignment and reduces computational cost.

The center of a protein complex with $N$ subdomains can be defined as follows.

$$C_p = M_p^{-1} \sum_{i=1}^{N} m_i c_i$$

where $C_p$ is the center of the protein complex with total mass $M_p$ and $c_i$ is the center of the $i$-th subdomain with mass $m_i$. For every projection $k$ the center $C_p^{(k)}$ can be written in terms of deviations of the mean positions of the N subdomains

$$C_p^{(k)} = M_p^{-1} \sum_{i=1}^{N} m_i \left( c_i + \Delta c_i^{(k)} \right)$$

where $C_p^{(k)}$ is the center of a protein complex for the ($k$-th)-projection; $\Delta c_i^{(k)}$ is a shift of the $i$-th subdomain from its center $c_i$. This shift reduces the accuracy of the registration (particle alignment) and eventually the resolution of the final 3D reconstruction. To solve this problem, (**Ilca et al., 2015**) focused the registration step on the $i$-th subdomain of the protein complex independently to find its particular shift and orientation. The proposed method extracts the projections of the $i$-th subdomain from the original stack (using for example the residual signal subtraction method of Bai et al., [**Bai et al., 2015**]) and stores them in a new stack with a new box size centered around the $i$-th subdomain. Reprojection of the center of the $i$-th subdomain, from the complete reconstructed complex (with a standard projection matching method) on to the ($k$-th)-projection, can be obtained by recentering the $i$-th subdomain.

$$c_i^{(k)} = proj_{2D}\left[ \left( R_p^{(k)} \right)^T \left( \hat{C}_p^{(k)} - c_i \right) \right]$$

where $c_i^{(k)}$ is the center of the i-th subdomain; $R_p^{(k)}$ is the orientation of the protein complex, $\hat{C}_p^{(k)}$ is its estimated center, for the ($k$-th)-projection and $c_i$ is the estimated center of the $i$-th subunit. The $proj_{2D}[]$ is an operator that projects 3D points to the XY plane.

The above procedure is implemented in a stand-alone program REP which reads STAR file formats as input that contain the alignment information derived from the consensus refinement, an MRCS file associated with the STAR file, and the subdomain centers from reconstruction $c_i$. REP generates a new STAR file with its corresponding MRCS file as output. The coordinates of the new center can be obtained from UCSF Chimera (**Pettersen et al., 2004**) or IMOD (**Kremer et al., 1996**). The alignment parameters, in a STAR file format, are automatically passed on to RELION (**Scheres, 2012**) or can be generated quickly using various cryo-EM software packages like EMAN2 (**Tang et al., 2007**), Frealign (**Grigorieff, 2007**) and Imagic (**van Heel et al., 1996**). The standalone program REP is written in C++ and can be downloaded from https://github.com/rkms86/REP (**Sanchez, 2017**). A copy is archived at https://github.com/elifesciences-publications/REP.

