## [Decision Letter]

Thank you for submitting your article "Cryo-EM structure of the bifunctional secretin complex of *Thermus thermophilus*" for consideration by *eLife*. Your article has been favorably evaluated by Richard Aldrich (Senior Editor) and three reviewers, one of whom, Wesley I Sundquist (Reviewer #1), is a member of our Board of Reviewing Editors.

The reviewers have discussed the reviews with one another and the Reviewing Editor has drafted this decision to help you prepare a revised submission.

The manuscript by D'Imprima et al. describes a cryoEM structure of a PilQ type IV pilus secretin complex from *Thermus thermophilus* at ~7 Å resolution. The resolution of the reconstruction was greatly improved using an in-house image processing tool, REP, that allowed the complex to be segmented into individual rigid components that were each processed separately. The *T. thermophilus* secretin complex has an especially large number of periplasmic domains, N0-N5, all of which are resolved in the EM map. Homology models of individual domains were generated and fit into the EM map. Linkers between several of the domain rings are well-resolved in the structure. A "crown" of density is apparent on the extracellular side of the complex that is not accounted for by the PilQ sequence. This density was not apparent in a PilQ variant with substitutions in two basic residues at the interface with the crown; this variant had a mild motility phenotype and a severe transformation phenotype suggesting that these residues and perhaps the crown itself are involved in DNA uptake. The secretin structure is of high quality and the results related to the crown are intriguing. Overall the work appears well done and paper is clearly and concisely written, but there are still some issues that need to be addressed before the manuscript will be acceptable for publication.

Major issues that need to be addressed:

1) A significant shortcoming is that the authors have not identified the protein that comprises the crown. This should be feasible by mass spec/proteomics approaches and seems necessary to complete the study, particularly as the information could substantially augment the existing structure and/or functional interpretation if the missing component proved to be homologous to another protein of known structure or function (e.g., a DNA binding protein). At a minimum, if this is not possible, then the authors should explain the lengths they have taken to identify this protein and why they have not been successful.

2) Data quality and assessments of the modeling/fitting. Firstly, the authors should explain the reason(s) that a high resolution structure could not be obtained. Secondly, docking at 7A resolution can be quite inaccurate and needs the maximum possible validation. Specifically, the authors need to provide more information regarding the accuracy of the homology models and how well they fit into the cryoEM maps (e.g., please provide percent similarities for the alignments used for homology modeling, statistical analyses of the fitting of domains into the maps, close-up/extracted images of protomers with more than one view for each of the domain fits so that readers can assess their quality, and apply a consistency test by generating a 7A map from the model and comparing it with the experimental map). For example, all published high resolution secretin structures show a continuous beta-stranded belt close to the outer membrane, but in the structure described here, the EM-density is quite discontinuous, which largely contrasts the proposed atomic model (Figure 2, Video 1, Figure 3).

Finally, the authors should also show how much the homology models diverged from the final models in cases where they were flexibly fitted into the EM map.

3) The C13 symmetry assignment needs to be solidified. Why were the eigenimages in Figure 1—figure supplement 1 performed on only a very small subset of particles (n=27)? What was the basis for selecting 27 out of 363 top-views (subsection “Cryo-EM image processing”, first paragraph). How does the (relevant) eigenimage appear if no random rotation is performed? It is expected that the second eigenimage will only show the 13-fold arrangement if all particles in top views are also not tilted. Can this be excluded? It is surprising that the 13-fold arrangement in the eigenimage (F) is observed at a level that is outside the ring. Could it be that the alternating pattern is a result of noise, tilted views, etc.? The authors should perform an eigenimage analysis on tightly ring-masked particles only, excluding all potential noise outside and inside the ring. Increasing the number of particles would also be of benefit.

4) The authors describe a novel recentering (REP) approach. Yet, how much does the result differ from Ilca et al. 2015? Did the authors do a comparative study?

---

## [Author Response]

Major issues that need to be addressed:1) A significant shortcoming is that the authors have not identified the protein that comprises the crown. This should be feasible by mass spec/proteomics approaches and seems necessary to complete the study, particularly as the information could substantially augment the existing structure and/or functional interpretation if the missing component proved to be homologous to another protein of known structure or function (e.g., a DNA binding protein). At a minimum, if this is not possible, then the authors should explain the lengths they have taken to identify this protein and why they have not been successful.

We fully agree with the reviewer that the protein that comprises the crown is of major interest. We found that the crown structure is essential for natural transformation and so we put a lot of effort into the identification of the crown protein. We started with a mass spectrometry approach of the purified PilQ complex. This led to the detection of more than 150 proteins in the purified PilQ preparations. Since the PilQ purification was performed with membrane extracts several well-known membrane associated/integrated proteins were detected, such as transporters, ATPases, the pyruvate DH complex, a cytochrome c oxidase, solute binding proteins, porins, lipoproteins and others. However, these studies did not reveal a good candidate for the crown protein.

The mass spectrometry analysis also led to the detection of 50 uncharacterized or hypothetical proteins. Several of them were closely associated with pilin-like genes or known genes of the natural transformation machinery and we speculate that one of these genes might encode the crown protein. We selected five of these uncharacterized/hypothetical genes, linked to pilins or genes of the transformation machinery and generated deletion mutants. However, none of these mutants were defective in natural transformation, which indicates that these proteins might not be directly involved in crown-mediated natural transformation.

We analyzed all proteins from *T. thermophilus* having sequence signatures resembling the *V. cholerae* GspD lip domain. We found one hit with a hypothetical membrane protein. This was also detected as part of a mass-spectrometry analysis of PilQ complex preparations. However, deletion of this candidate gene did not affect transformation, which indicates that this protein is not involved directly in DNA binding or natural transformation mediated by the crown.

2) Data quality and assessments of the modeling/fitting. Firstly, the authors should explain the reason(s) that a high resolution structure could not be obtained.

The organization of seven individual PilQ domains into several rigid ring structures connected by flexible linkers segments makes this complex highly dynamic. The increased flexibility enhances inter-ring motions, resulting in several dynamic states. The stabilization of these states depends on several external factors such as buffer conditions/temperature and imaging parameters. Our approach of reconstructing of individual rigid modules improved the resolution dramatically from 20 to 7 Å, and an even bigger improvement could not be expected.

Secondly, docking at 7A resolution can be quite inaccurate and needs the maximum possible validation. Specifically, the authors need to provide more information regarding the accuracy of the homology models and how well they fit into the cryoEM maps (e.g., please provide percent similarities for the alignments used for homology modeling, statistical analyses of the fitting of domains into the maps, close-up/extracted images of protomers with more than one view for each of the domain fits so that readers can assess their quality, and apply a consistency test by generating a 7A map from the model and comparing it with the experimental map). For example, all published high resolution secretin structures show a continuous beta-stranded belt close to the outer membrane, but in the structure described here, the EM-density is quite discontinuous, which largely contrasts the proposed atomic model (Figure 2, Video 1, Figure 3).

We are aware of the challenges in docking/fitting homology models into EM maps, both because of the quality of the structural models and because of the resolution of the map. To address this issue, we now (i) list percent of the similarities of the alignments used for homology modelling, (ii) provide detailed statistical analysis of the fitting of domains into the maps, and (iii) show closer views of the models in the maps.

To assess the statistics of the fits, we compared the cross-correlation coefficient (CC) after fitting of PilQ domains to the distribution of the CCs obtained in extensive sampling of orientations of the initial model (*n* = 10000; complete ensemble of fits). We show that the best fitting orientations in this initial search rank within the top five best solutions from the complete ensemble of fits. This implies that the initial best fit orientations had good coverage. To assess convergence, we checked that repeated optimizations produced identical best-fit orientations. We summarize the results of fitting procedures by showing the cumulative distribution of the entire ensemble of fits for each domain along with the CC-value of its initial chosen best fit before and after refinement using MDFF approach. We have added this information to the Materials and methods and also added two new figures (Figure 3—figure supplement 4, Figure 3—figure supplement 5) and a new Table 2 summarizing these results. We also provide close-up images of initial and refined fits in multiple orientations for all PilQ domains (Figure 3—figure supplement 6). Further we have also added a video (Video 2) showing the final atomic model fit into the density maps. The movie displays individual domain fits in close-up by sectioning through the various ring modules of the entire PilQ complex.

Modelling details:

Table 1 (modified now) also provides information about the homology modelling procedure of individual domain models. It clearly provides information about individual domain templates, their sequence identities/similarities with the target PilQ domain along with individual DOPE scores/Z-scores.

Model/map validation:

The initial best fitted C-13 homology models before refinement were used to generate model derived EM maps at the same nominal resolution as experimental recentered EM maps. Comparison of model derived and experimental EM maps was performed by computing per voxel local cross-correlation (using the *vop localCorrelation* module of UCSF Chimera) to show model/map correspondence. This showed an overall good agreement between the initial C-13 symmetric homology models and the EM maps. Regions that were not resolved in original EM maps, i.e. corresponding to linker segments and domain termini connecting individual maps were subsequently refined using MDFF.

Discontinuity of beta-barrel density:

All published high resolution structures of Secretins display a continuous beta-belt density corresponding to the outer beta barrel. Beta sheets are only resolved in to individual strand at resolution higher than 4.7Å and are hard to interpret at lower resolution. The discontinuous density in the beta-barrel region of our map (Figure 2 and Figure 3) is highly dependent on the contour level selection. In Figure 2 and Figure 3, the contour level of the map was chosen to highlight features of the crown and N5 regions, which would otherwise not be clear. The beta barrel-region has a continuous density when visualized at a lower contour level. Therefore, our homology model, which is quite similar to other well characterized secretins (Figure 4), is compatible with the current map. We have made an additional figure (Author response image 1) to highlight the continuity/discontinuity as a function of the contour level.

Finally, the authors should also show how much the homology models diverged from the final models in cases where they were flexibly fitted into the EM map.

Structural divergence after MDFF:

MDFF refinement adds forces derived from density gradients in the EM maps to a molecular simulation model. Constraints on the secondary structure and chirality of residues ensure that the model structure is changed minimally to avoid overfitting. The variation of the structure after refinement is therefore a measure of the model divergence required to obtain a satisfactory fit (better/improved CC). We find that the average Cα-RMSD for individual domain models before and after MDFF is ~2.7 Å, indicating that the divergence of domain structures during refinement procedure was minimal and restricted to long loops and termini only. The largest change was observed for the N1 domain during refinement (4.04 Å), which contains several long loops connecting individual secondary structure elements. We have added another column to Table 2 providing this information.

3) The C13 symmetry assignment needs to be solidified. Why were the eigenimages in Figure 1—figure supplement 1 performed on only a very small subset of particles (n=27)? What was the basis for selecting 27 out of 363 top-views (subsection “Cryo-EM image processing”, first paragraph).

Out of 363 top-views 27 where chosen because they seem not to be tilted in respect to the imaging plane, therefore ensuring the interpretability of the eigenimages.

How does the (relevant) eigenimage appear if no random rotation is performed?

The results are consistent. Multiple copies randomly rotated are generated in order to get more robust statistics, see Böttcher et al., Structure 23:1705-14, 2015.

It is expected that the second eigenimage will only show the 13-fold arrangement if all particles in top views are also not tilted. Can this be excluded?

The reviewer is correct, for this reason we selected only the 27 particles that look untilted. The fact that we do detect 13-fold symmetry shows that this assumption was mostly correct.

It is surprising that the 13-fold arrangement in the eigenimage (F) is observed at a level that is outside the ring. Could it be that the alternating pattern is a result of noise, tilted views, etc.? The authors should perform an eigenimage analysis on tightly ring-masked particles only, excluding all potential noise outside and inside the ring. Increasing the number of particles would also be of benefit.

We made sure that the mask which we used to calculate the principal components was not too tight and supply a figure to show this (Author response image 2). The circular mask applied during 2D classification includes some noise surrounding the particle images (A). However, the binary annular mask used during multivariate statistical analysis (B) which in (C) is multiplied by the class average clearly show that it excludes the noise outside the particle ring and just includes the symmetric spikes radiating from the particle top views. The same mask is shown multiplied by the second eigenimage (D) which corresponds to Figure 1—figure supplement 1.

The fact that ultimately we obtained a three-dimensional structure showing secondary structure validates our assessment of the structure as 13-fold symmetric.

**Author response image 2. respfig2:** Assessment of PilQ symmetry and oligomeric state. (**A**) 2D class average derived from 27 top views was not sufficient to assess the symmetry unambiguously. (**B**) Binary annular mask applied for multivariate statistical analysis. (**C**) Multiplication of (**A**) by (**B**) shows that the applied mask is not too lose and it just includes signal from the ring densities. (**D**) Second eigenimage after multivariate statistical analysis.

4) The authors describe a novel recentering (REP) approach. Yet, how much does the result differ from Ilca et al. 2015? Did the authors do a comparative study?

Our idea was to provide a method which is simple, flexible, modular and fast.

The differences between REP and the program from Ilca et al. 2015 relates to two core points: what the program actually does (1) and how it operates (2).

(1) Ilca's localized reconstruction tool provides a final refinement step to a specific area of the 3D map. Its general procedure consists of four steps: a initialization step which includes preparation of the necessary metadata for the localized reconstruction and the Residual Signal Substraction (RSS) procedure. This is followed by extraction of the subparticles and a final reconstruction with refinement.

On the other hand, our implementation, REP, is designed as a separate processing step of the whole reconstruction procedure. We only compute the shifts and orientations of the subparticles followed by extraction. This way we can apply the recentering procedure to the data whether this has been already processed with RSS or not. The modularity of REP allows extra flexibility, as the subtraction procedure and final refinement steps are optional and can be performed by any processing software package.

(2) The current implementation of Ilca's localized reconstruction uses Python as programming language, and it requires the installation of the Scipion and Relion-1.4 Python libraries.

We think that this is a drawback, as it forces the user to adopt specific programs which may not even be compatible with newer versions. With this in mind, we decided to implement REP using C++ and without using any additional library. By doing this we ensure that we can always use the state-of-the-art algorithms for signal subtraction, reconstruction and refinement provided by any image processing package. The concept of recentering is not a novelty per-se. The corresponding author of Ilca et al. 2015 was for example already co-author of one of the first examples, to our knowledge, of a similar approach (Briggs et al., J Struct Biol.150:332-9, 2005).

We did not do a comparative study since the program from Ilca et al. 2015 makes use of the same concept of particle recentering. The main novelty of our approach is that REP is a stand-alone program completely independent from any other reconstruction software.